# 24-Hour Rest-Activity Rhythm in Middle-Aged and Older Persons with Depression

**DOI:** 10.3390/ijerph20075275

**Published:** 2023-03-27

**Authors:** Masaki Hayashi, Masahiro Takeshima, Tomoko Hosoya, Yu Kume

**Affiliations:** 1Graduate of School of Health Sciences, Akita University, Akita 010-8543, Japan; 2Department of Neuropsychiatry, Graduate School of Medicine, Akita University, Akita 010-8543, Japan; 3Department of Occupational Therapy, Graduate School of Medicine, Akita University, Akita 010-8543, Japan

**Keywords:** depression, rest-activity, actigraph

## Abstract

Changes in rest or active states were clinically observed in persons with depression. However, the association between symptoms of depression and 24 h rest-activity rhythm (RAR) components that can be measured using wearable devices was not clarified. This preliminary cross-sectional study aimed to clarify the 24 h RAR components associated with symptoms of depression in middle-aged and older persons. Participants were recruited from among inpatients and outpatients requiring medical treatment at Akita University Hospital for the group with depression and from among healthy volunteers living in Akita prefecture, Japan, for the healthy control group. To assess RAR parameters including inter-daily stability (IS), intra-daily variability (IV), relative amplitude (RA), and average physical activity level for the most active 10 h span (M10) or for the least active 5 h span (L5), all the participants were instructed to wear an Actiwatch Spectrum Plus device on their non-dominant wrist for seven days. Twenty-nine persons with depression and 30 controls were included in the analysis. The results of a binomial regression analysis showed that symptoms of depression were significantly associated with a high IS value (odds ratio [OR], 1.20; 95% confidence interval [95% CI], 1.01–1.44; *p* = 0.04) and a low M10 value (OR, 0.85; 95% CI, 0.74–0.96; *p* = 0.01). Our findings suggest potential components of 24 h RAR are associated with depression.

## 1. Introduction

A high prevalence of affective disorder is commonly recognized worldwide; in particular, the number of patients with depression was estimated to have reached 3.8% of the global population (approximately 280,000,000 people) [1]. In the Japanese population, the estimated numbers of inpatients and outpatients with depression in 2020 were 280,000 and 910,000 patients, respectively [2]. For those with depression, non-invasive or non-pharmacological therapeutic approaches such as exercise therapy [3], cognitive behavior therapy [4], and psychoeducation [5] are recommended clinically. Therapeutical approaches targeting specific symptoms of depression, such as night-sleep disorder and low physical activity levels in daily life, were well documented in the last few decades; one study suggested that patients with depression and a low daily physical activity level might have significantly shorter night-sleep times than healthy controls, as well as a significantly low relative amplitude (RA) between physical activity levels during daytime and those during nighttime [6]. In a cross-sectional study targeting community-dwellers with mild to moderate depression and anxiety, longer sedentary times during 24 h daily periods were observed [7], and longer screen-times associated with sedentary behaviors were reported in a systematic review [8]. Moreover, a deterioration in 24 h rest-activity rhythm (RAR) during the daytime was considered a risk factor for symptoms of depression in middle-aged and older persons [9,10], with one reference describing circadian rhythm or sleep alterations in older persons with lifetime depression [11]. Given this collective background, symptoms of depression in middle-aged and older persons might impact changes in 24 h RAR patterns in the subjects’ daily lives. However, limited information is available regarding changes in 24 h RAR in middle-aged and older persons with depression and, especially, the characteristics of 24 h RAR in patients receiving medical treatment for depression remain controversial because of the applications of different scales for evaluating depression within similar research fields.

The objective of the present study was to examine the 24 h RAR components associated with symptoms of depression on a preliminary basis in middle-aged and older persons. To clarify the 24 h RAR components in middle-aged and older persons with depression, this study focused on the following points: (i) to clarify differences in the RAR measured using an accelerometer between patients with depression and healthy controls, (ii) to analyze the correlation between rest-activity parameters and social skills in patients with depression, (iii) to extract factors associated with rest-activity parameters suggestive of the presence of depression, and (iv) to obtain a general view of 24 h RAR components in patients with depression and healthy controls. With the main point of association between 24 h RAR components and depression which would be analyzed by the binomial logistic regression model, we hypothesized that 24 h RAR components related to rhythm stability and states of physical activity or rest as measured using an accelerometer in daily life would be identified in middle-aged and older persons with depression. These potential findings would provide clinicians with meaningful viewpoints for determining treatment or care strategies for patients with depression.

## 2. Materials and Methods

### 2.1. Participants

The presently reported cross-sectional observational study was performed from January 2022 to February 2023. Participants were recruited from among inpatients or outpatients receiving treatment at Akita University Hospital or community-dwellers living in Akita prefecture, Japan. The recruitment of inpatients and outpatients was performed using a poster in the hospital, and the psychiatrist in charge of each patient obtained the patient’s informed consent to participate in this study. Healthy persons living in the community were recruited through a public bulletin in Akita prefecture. Informed consent was obtained from all community participants. The inclusion criteria were as follows: (i) an age of 18 years old or older, (ii) a diagnosis of an affective disorder according to the Diagnostic and Statistical Manual of Mental Disorders-V (DSM-V), and (iii) inpatients or outpatients with a score of 8 or more of the Hamilton Depression Rating Scale-17 (HDRS-17). The exclusion criteria were (i) a strong refusal to wear a wearable device, (ii) severe psychoneurotic symptoms, (iii) central nervous system disease requiring supportive care for daily life activities, and (iv) a score of 5 or more on the Geriatric Depression Scale-15 (GDS-15) within the control group. 

Demographic information was collected from electronic medical records at Akita University Hospital; information included age, sex, education, days of hospitalization, onset, presence of readmission, living with family, history of divorce, medication (antipsychotic or benzodiazepine), and independence of daily living activities according to the Barthel index. Demographic data was included in each subject’s medical record at Akita University Hospital after the completion of an interview by the psychiatrist in charge.

### 2.2. Measurement

#### 2.2.1. Depression Scale

The Hamilton Depression Rating Scale-17 (HDRS-17) was used in this study [12]. Items of HDRS-17 comprised of (1) depressed mood (sadness, hopeless, helpless, worthless) (a score of 0 to 4), (2) feelings of guilt (a score of 0 to 4), (3) suicide (a score of 0 to 4), (4) insomnia: early in the night (a score of 0 to 2), (5) insomnia: middle of the night (a score of 0 to 2), (6) insomnia: early hours of the morning (a score of 0 to 2), (7) work and activities (a score of 0 to 4), (8) intellectual disability (slowness of thought and speech, impaired ability to concentrate, decreased motor activity) (a score of 0 to 4), (9) agitation (a score of 0 to 4), (10) anxiety psychic (a score of 0 to 4), (11) anxiety somatic (physiological concomitants of anxiety) (a score of 0 to 4), (12) somatic symptoms gastro-intestinal (a score of 0 to 2), (13) general somatic symptoms (a score of 0 to 2), (14) genital symptoms (symptoms such as loss of libido, menstrual disturbances) (a score of 0 to 2), (15) hypochondriasis (a score of 0 to 4), (16) loss of weight (rate either [a] or [b]) (a score of 0 to 3 [a] according to the patient or [b] according to weekly measurements), and (17) insight (a score of 0 to 2) [12]. Trained psychiatrists at Akita University Hospital applied the HDRS-17 according to a structured interview guide for the HDRS [13]. For the HDRS-17, a score of 0 to 7 falls within the normal range, while a score of 20 or higher indicates at least moderately severe depression [14]. 

#### 2.2.2. Non-Parametric Circadian Rest-Activity Parameters

The participants wore an Actiwatch Spectrum Plus (AW-SP) (Philips Respironics, Inc.) on their non-dominant wrist for 7 continuous days so as to measure nonparametric rest-activity rhythm (RAR) parameters. AW-SP data (activity counts, AC) were collected at one-minute intervals and analyzed using Actiware version 6.30 (Philips Respironics, Inc.). The RAR parameters consisted of inter-daily stability (IS), intra-daily variability (IV), and relative amplitude (RA), with the original references [15,16]. The IS values ranged from 0 for a normal distribution wave to 1 for a completely stable rhythm against environmental or external stimuli. The IV values showed a fragmented activity pattern, ranging from 0 for RAR without fragmentation to 2 for fragmented RAR. The RA values reflected the relative proportion between the average AC during the most active 10 h span (M10) and the average AC during the least active 5 h span (L5) in the 24 h activity pattern, ranging from 0 for a low RA between M10 and L5 to 1 for a high RA between M10 and L5.

#### 2.2.3. Social Functioning Assessment

The University of California San Diego, Performance-Based Skills Assessment-Brief (UPSA-B) consisted of two subscale scores, finances and communication, each of which was converted into a standard score ranging from 0 to 50 points; thus, the maximum total score was 100 points. The first subscale, finances, consisted of tasks related to counting money, calculating change, and paying bills; the second subscale, communication, consisted of role-playing tasks using props (money, bills, letters, phone calls, etc.) with the themes of using the telephone and changing medical examination appointments. Social life skills (functional abilities) in daily life were graded according to the achievement level of each task. The cutoff value of the Japanese version of the UPSA-B is 80 points [17,18,19].

#### 2.2.4. Data Analysis

First, an unpaired *t*-test or Mann–Whitney U test was performed to compare demographic data and RAR parameters between the group with depression and the healthy group, depending on the confirmation of a normal distribution. Nominal scales, such as sex, were analyzed using the chi-square test. Second, the Spearman’s rank correlation coefficient test was applied to examine correlations between each RAR parameter and other clinical variables within the group with depression. Third, for a binominal logistic regression analysis, the likelihood ratio test with forward selection was performed to extract factors associated with the 24 h RAR component in middle-aged and older persons with depression. A dependent variable of the group classification (dummy variable, healthy group = 0; depression group = 1) was created for the binominal logistic regression analysis and likelihood ratio test, and independent variables were selected statistically for the crude regression model according to the results of the univariate or bivariate analyses with a reference value of *p* < 0.25. The final model was decided taking into consideration the results of model adaptation using the Hosmer–Lemeshow test, the significant variables in each model, and the percentage of correct classifications and R square (Nagelkerke’s R^2^) value for the contribution rate of each model. To avoid an extremely high or low odds ratio (OR) for each variable in the regression model [20], the numerical conversion of ×100 or ×1/100 was additionally performed for each RAR parameter [21]. Lastly, the average activity plots were created to allow visual comparisons between groups and were analyzed using the Mann–Whitney U test. The statistical analysis was performed using SPSS Version 27.0 for Windows (SPSS Inc., Chicago, IL, USA), and the level of significance was set at *p* = 0.05.

## 3. Results

### 3.1. Demographic Data

Table 1 lists the demographic data for the healthy group (n = 30) and the depression group (n = 29; inpatients, n = 24; outpatients, n = 5) and the results of the univariate analyses (unpaired *t*-test, Mann–Whitney U test, or chi-square test). For the depression group, the average ± standard deviation (SD) of age was 53.7 ± 17.4 years, and the ratio of females (%female) was 46.7%; in the healthy group, the average age ± SD was 53.1 ± 18.0 years and the %female was 50.0%. Age and sex were not significantly different between the groups (*p* > 0.05). Within the depression group, the median (interquartile range [IQR]) of the HDRS-17 was 18.0 [18.0] points, and the severities of symptoms of depression were categorized as mild (n = 6), moderate (n = 13), severe (n = 2), or very severe (n = 8). Additionally, the median (IQR) of the UPSA-B total score was 61.0 (27.0) points, and the UPSA-B subscales for finance and communication were 41.0 (7.0) and 16.0 (23.0) points, respectively. According to the results of Mann–Whitney U tests, the values for education, L5 and M10, were significantly lower in the depression group than in the healthy group (Table 1), and the depression group had a larger IV than the healthy group (Table 1). Other RAR variables (IS and RA) were not significantly different between the groups (*p* > 0.05).

### 3.2. Correlation in the Group with Depression

Table 2 shows the results of the Spearman’s rank correlation coefficient test for the depression group (n = 29). Age was negatively correlated with education (r = −0.55, *p* < 0.01). The IS variable was negatively correlated with the IV variable (r = −0.56, *p* < 0.01) and was positively correlated with the M10 value (r = 0.45, *p* < 0.05). Moreover, the L5 value was negatively correlated with the RA variable (r = −0.87, *p* < 0.01), as well as being positively correlated with the M10 value (r = 0.61, *p* < 0.01). However, a significant correlation between the UPSA-B and RAR variables was not observed in the depression group (*p* > 0.05).

### 3.3. Binomial Regression Model

Based on the results of the above-mentioned univariate (Table 1) and bivariate (Table 2) analyses, the independent variables of age, IS × 100, IV × 100, RA × 100, L5/100, and M10/100, were inputted into a binominal logistic regression model using the group classification (dummy variable: healthy group = 0; depression group = 1) as the dependent variable. Table 3 lists the estimated regression models. For the crude model, the classification of depression was significantly associated with M10×1/100 (OR, 0.92; 95% confidence interval [95% CI], 0.88–0.96; *p* = 0.001), with a good model adaption (Hosmer–Lemeshow test, *p* = 0.74 > 0.05) and 0.832 of Nagelkerke’s R^2^. Next, Model I showed that the group with depression was significantly associated with IS × 100 (OR, 1.20; 95% CI, 1.01–1.44; *p* = 0.04) and M10 × 1/100 (OR, 0.85; 95% CI, 0.74–0.96; *p* = 0.01), with a better goodness-of-fit with the model (Hosmer–Lemeshow test, *p* = 0.74 > 0.05) and 0.902 of Nagelkerke’s R^2^. Finally, Model II had no significantly associated factors after adjustments for age (Table 3). Although each model had a high discrimination accuracy of 93.2%, Model I was selected as the best model in this study because of the significant association of RAR variables (IS × 100 and M10 × 1/100).

### 3.4. Activity-Plots for Each Group

Figure 1 shows the 24 h activity-plots for each group. For the group with depression, the peak AC was 8180 AC at timeline 17 (PM 16:00–16:59) and the lowest AC was 531 AC at timeline 1 (AM 0:00–0:59). For the healthy group, the highest AC was 18695 AC at timeline 8 (AM 7:00–7:59) and the lowest AC was 1234 AC at timeline 4 (AM 3:00–3:59). A comparison test indicated that timeline 1 (AM 0:00–0:59) and timelines 7 to 24 (AM 6:00-PM 23:59) were associated with significantly lower AC values in the group with depression than in the healthy group (Figure 1).

## 4. Discussion

As the main point of this study, the results of binomial logistic regression analysis suggest that symptoms of depression in middle-aged and older persons might be significantly associated with a stable rhythm and low levels of most active spans over a 24 h period. However, correlation between 24 h RAR components and social functioning skill as indexed in UPSA-B was not observed in the persons with depression.

Regarding the logistic regression model (Model I) produced in this study, a cross-sectional actigraphy study previously indicated that, in an estimated model with adjustments for age and sex, an unstable rhythm as indexed by a low IS value was associated with an increase in symptoms of depression (beta = −0.05, standard error [SE] = 0.02, *p* = 0.047) in middle-aged and older persons (average age ± SD, 62.2 ± 9.4) [9]. A report by Luik et al. used the Center for Epidemiologic Studies-Depression (CES-D) scale to evaluate symptoms of depression in subjects; this methodology differed from the criteria used in this study for participants with depression requiring inpatient (n = 24) or outpatient (n = 5) treatment [9]. As the lifestyles of the inpatients in the present study might have been influenced by the typical time schedule of Japanese general hospitals, including the regular hours of waking, bedtime, and meals, the RAR stability of these patients might have been affected by their environment. Moreover, we cannot unconditionally deny the effects of behavior restrictions on either inpatients or outpatients, as this study was conducted during the COVID-19 pandemic in Japan.

The M10 variable of the RAR components was extracted as an associated factor in Model I. The M10 variable reflects the active span of physical activity over a 24 h time period, and an association between a low level of physical activity and symptoms of depression was previously reported: Helgadottir et al. (2015) reported that people affected by depression and anxiety disorders had a sedentary lifestyle and completed their daily activities within a short period of time, spending approximately 9.1 h awake [7]. Difrancesco et al. (2019) focused their research on the intensity of physical activity and the severity of depression, suggesting that a decrease in physical activity with a moderate intensity of 3.0 to 5.9 metabolic equivalents (METs) was significantly associated with severe symptoms of depression (beta = 0.24; SE = 0.05, *p* < 0.001) [6]. Although these findings cannot be directly compared with the low M10 values seen in Model I in the present study, a low level of physical activity during the day in middle-aged and older persons with depression was suggested by the results of our study. Since trained psychiatrists additionally applied the DSM-V and the HDRS to diagnose affective disorder in the participants, our observational study might have had an impact on the low physical activity level of middle-aged and older persons requiring medical treatment.

The activity plots in Figure 1 showed a significantly lower level of physical activity in the group with depression, compared with the healthy controls. Previous studies supporting our findings were not available, but some chronological observational studies of physical activity in daily life reported relevant information. For example, Burton et al. (2013) performed a systematic review (including 19 articles regarding observational actigraph studies) and reported that persons with depression (controls, n = 208; persons with depression, n = 245; random effect analysis) exhibited a significant decline in physical activity during the daytime (standardized mean difference, −0.76; 95% CI, −1.05, −0.47), with a moderate heterogeneity (I-squared = 47.5%) [22]. A case–control study (10 controls vs. 10 cases with depression) in Japan reported by Ueda et al. (2005) suggested that physical activity within the span of 12:00 to 18:00 in persons with depression was higher than that in a control group, while physical activity within the span of 18:00 to 00:00 was significantly lower in persons with depression, compared with the control group [23]. In contrast to previous findings regarding physical activity in daytime, relatively high physical activity within a span of 12:00 to 18:00 was observed potentially in patients with the early period of depression [24]. As indicated in Figure 1, physical activity peaked during timeline 17 (16:00 to 16:59; average AC = 8180 AC) over the 24 h time period in the group with depression [24]. A meta-analysis of five articles examining sedentary lifestyles in persons with depression revealed extended sedentary times (standardized mean difference, 0.09; 95% CI, 0.01, 0.18; *p* = 0.02) [24]. In the present study, the peak point in physical activity or sedentary lifestyle during the daytime in middle-aged and older persons with depression was unclear, but further examination is warranted.

An additional analysis examining the correlation between RAR variables and social functioning skills within the group with depression did not produce any significant results (Table 2). Among the few reports on physical activity and social functioning in depression, Cohn-Schwartz et al. (2022) recently reported that, in a multiple mediation model analysis, the intensity of physical activity during daily life was associated with a high level of cognitive performance (beta = 0.17; *p* = 0.007), and its association was significantly mediated by the outcome of symptoms of depression (beta = 0.009; 95% CI, 0.009, 0.025) and social interactions (beta = 0.022; 95% CI, 0.005, 0.046) [25]. In addition, the comparisons shown in Table 1 indicated that IV values in the depression group were significantly higher than those in the healthy group. Although IV was not extracted as an associated factor in the regression model, fragmented RAR patterns in middle-aged and older persons were well documented in population-based surveys. Luik et al. (2013) suggested that symptoms of depression in middle-aged and older persons were associated with the fragmentation of RAR variables (beta = 0.10; *p* < 0.001) [26]. Among other RAR components, the association between RA, which reflects the reflected relative proportion of rest and activity states during the day, and symptoms of depression or anxiety remains controversial [6].

Finally, some limitations of our study should be mentioned. First, our study samples were small, with a bias toward inpatients (inpatients, n = 24; outpatients, n = 5) within the group with depression. As mentioned previously, in-hospital medical treatment and hospital routines likely influenced the activity levels of the inpatients with depression. Meanwhile, outpatients with an HDRS score of more than eight were difficult to recruit, as well as control of demographic information (education, days of hospitalization, onset, presence of readmission, et al.) Secondly, our study was conducted at Akita University Hospital during the COVID-19 pandemic in 2022, and social behavior restrictions might have had a negative influence on lifestyle behaviors during this time period. A final limitation was a bias towards a higher level of social functioning in persons with depression, as measured by the UPSA-B. Although social functioning assessments of the healthy controls were not included in this study design, future studies should examine whether social functioning skills influence physical activity levels during daily life.

## 5. Conclusions

The present study aimed to clarify the 24 h RAR components associated with symptoms of depression on a preliminary basis in middle-aged and older persons. Our results indicated that symptoms of depression in middle-aged and older persons were associated with the stability of 24 h RAR and a low level of physical activity during the most active span over a 24 h time period. However, correlation between 24 h RAR components and social functioning skill was not revealed in middle-aged and older persons with depression. Moreover, the potential impact of chronological changes in physical activity patterns, as revealed by activity plots over 24 h, on symptoms in middle-aged and older persons with depression cannot be ruled out and warrant further examination.

## Figures and Tables

**Figure 1 ijerph-20-05275-f001:**
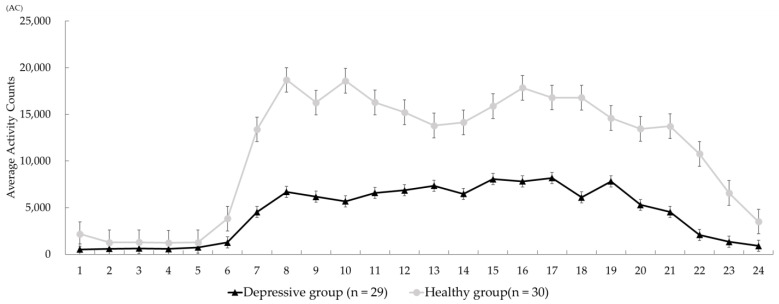
Comparison of activity-plots between the groups. Each error bar indicates the standard error of the mean (SEM) according to the hourly average of activity counts. The number of horizontal axis (Timeline) shows along the axis of time from 0:00 to 23:59 (e.g., the number of 1 means 0:00–0:59 and the number of 24 indicates 23:00–23:59). AC, activity counts.

**Table 1 ijerph-20-05275-t001:** Demographic data.

**Variable**	**Healthy Group** **(n = 30)**		**Depressive Group** **(n = 29)**		* **p** * **Value**
	**Mean**	**SD**	**Mean**	**SD**	
**Age (year)**	53.1	18.0	53.7	17.4	0.89
**Gender (Female/Male, n)**	15/15		14/15		0.37
**Hospitalization days (days)**	-	-	24.3	18.7	-
**Onset years (years)**	-	-	11.8	9.9	-
**Readmission (Yes/No, n)**	-		15/14		-
**Living with family (Yes/No, n)**	-		20/9		-
**Spouse (Yes/No, n)**	-		20/9		-
**Antipsychotic (Yes/No, n)**	-		18/11		-
**Benzodiazepine (Yes/No, n)**	-		10/19		-
**Variable**	**Median**	**IQR**	**Median**	**IQR**	***p* value**
**Education (year)**	16.0	5.0	14.0	4.0	<0.01
**Barthel Index (score)**	100.0	0	100	8.0	-
**HAMD (score)**	-	-	18.0	18.0	-
**Severity of HAMD (mild/moderate/severe/very severe, n)**	-	-	6/13/2/8		-
**UPSA-B total score (score)**	-	-	61.0	27.0	-
**UPSA-B communication (score)**	-	-	16.0	23.0	-
**UPSA-B finance (score)**	-	-	41.0	7.0	-
**Wearing time of AWSP (hour)**	173.0	61.0	173.0	16.0	0.93
**IS (range, 0.00–1.00)**	0.57	0.15	0.55	0.17	0.34
**IV (range, 0.00–2.00)**	0.91	0.32	1.28	0.38	<0.01
**RA (range, 0.00–1.00)**	0.91	0.07	0.90	0.09	0.16
**L5 (count)**	764.6	603.6	458.3	536.3	<0.01
**M10 (count)**	16,142.1	5288.0	7295.8	4380.0	<0.01

The *p* value within Table 1 means a statistical result analyzed by the unpaired *t*-test, the chi-square test, or the Mann–Whitney U test for nominal variables between the groups. SD, standard deviation; IQR, interquartile range; HAMD, Hamilton Rating Scale for Depression; UPSA-B, University of California San Diego, Performance-Based Skills Assessment-Brief; AWSP, Actiwatch Spectrum Plus; IS, inter-daily stability; IV, intra-daily variability; RA, relative amplitude; L5, average activity counts average of the least active 5 h period calculated by the average 24 h profile; M10, Average activity counts of the most active 10 h period calculated by the average 24 h profile.

**Table 2 ijerph-20-05275-t002:** Correlation between parameters in the group with depressive state (n = 29).

	Median	IQR	1	2	3	4	5	6	7	8	9
**1. Age**	56.0	32.0									
**2. Education**	14.0	4.0	−0.55 **								
**3. BI**	100.0	8.0	−0.34	0.26							
**4. HAMD**	18.0	18.0	0.11	−0.22	0.15						
**5. UPSA-B**	61.0	27.0	−0.37	0.43 *	−0.05	0.00					
**6. IS**	0.55	0.17	0.09	−0.06	0.00	0.00	0.11				
**7. IV**	1.28	0.38	0.15	−0.14	−0.26	0.10	−0.23	−0.56 **			
**8. RA**	0.90	0.09	0.34	−0.20	−0.03	0.10	−0.21	0.05	0.22		
**9. L5**	458.3	536.3	−0.22	0.09	0.01	0.00	0.12	0.19	−0.27	−0.87 **	
**10. M10**	7295.8	4380.0	0.05	−0.04	−0.07	0.12	0.02	0.45 *	−0.24	−0.19	0.61 **

* *p* < 0.05, ** *p* < 0.01, Spearman rank correlation coefficient. The values in Table 2 indicate Spearman rank correlation coefficient between parameters. IQR, interquartile range; BI, Barthel index; HAMD, Hamilton Rating Scale for Depression; UPSA-B, University of California San Diego, Performance-Based Skills Assessment-Brief; IS, inter-daily stability; IV intra-daily variability; RA, relative amplitude; L5, average activity counts average of the least active 5 h period calculated by the average 24 h profile; M10, average activity counts of the most active 10 h period calculated by the average 24 h profile.

**Table 3 ijerph-20-05275-t003:** Binomial logistic regression models analyzed by the likelihood ratio test with forward selection.

	β	SE	Odds Ratio	95%CI		*p* Value
**Crude model**						
**M10 × 1/100 (counts)**	−0.08	0.02	0.92	0.88	0.96	0.001 **
**Constant**	10.26	3.14				0.001 **
**Model I**						
**IS value × 100**	0.19	0.09	1.20	1.01	1.44	0.04 *
**M10 × 1/100 (counts)**	−0.17	0.07	0.85	0.74	0.96	0.01 *
**Constant**	9.68	4.37				0.03 *
**Model II**						
**Age (years)**	−0.47	0.38	0.63	0.30	1.32	0.22
**IS value ×100**	1.32	1.05	3.75	0.48	29.14	0.21
**M10 × 1/100 (counts)**	−1.09	0.86	0.34	0.06	1.82	0.21
**Constant**	84.09	67.50				0.21

* *p* < 0.05, ** *p* < 0.01. Reference group for the analysis was the healthy group (e.g., healthy group = 0; depressive group = 1 for each category of dependent variables). Independent variables included age, sex, education, IS × 100, IV × 100, RA × 100, L5 × 1/100 and M10 × 1/100. Crude model: Model χ^2^ test, *p* < 0.001, Hosmer and Lemeshow’s test, *p* = 0.74, Nagelkerke’s R^2^ = 0.832 and discrimination accuracy, 93.2%. Model I: Model χ^2^ test, *p* < 0.001, Hosmer and Lemeshow’s test, *p* = 0.99, Nagelkerke’s R^2^ = 0.902 and discrimination accuracy, 93.2%. Model II: Model χ^2^ test, *p* < 0.001, Hosmer and Lemeshow’s test, *p* = 0.89, Nagelkerke’s R^2^ = 0.955 and discrimination accuracy, 93.2%. β, coefficient; SE, standard error; 95%CI, confidence interval; IS, inter-daily stability; IV, intra-daily variability; RA, relative amplitude; L5, average activity counts average of the least active 5 h period calculated by the average 24 h profile; M10, average activity counts of the most active 10 h period calculated by the average 24 h profile.

## Data Availability

No publicly archived datasets, analyzed or generated, were used in this study.

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
