# Peer review of "24-Hour Rest-Activity Rhythm in Middle-Aged and Older Persons with Depression"

_ijerph, 2023, doi:10.3390/ijerph20075275_

Round 1

Reviewer 1 Report

THe research is a valauable one. However, try to improve the main method and it's introduction in a sensible manner. Some essential components of regresion analysis have been missing.

Author Response

Dear Reviewer1,

We would like to appreciate your supportive comments toward our paper. According to suggestion raised by each reviewer, our manuscript has been revised carefully as indicated in the following response to each reviewer’s comment. We believe the revised paper would provide novel findings to the readers of International Journal of Environmental Research and Public Health with Special Issue "Rehabilitation in Geriatric Psychiatry".

Reviewer 1

The research is a valuable one. However, try to improve the main method and it's introduction in a sensible manner. Some essential components of regression analysis have been missing.

Comment 1: 24 rest activity rhythm has not adequately explained.

Response to Comment 1: Thank you for a supportive comment. Explanation of 24-hour rest-activity rhythm (RAR) has been added briefly in the abstract (page 1, abstract).

Comment 2: More empirical previous studies can be integrated in order to develop a powerful impact for the present study.

Response to Comment 2: Thank you for a supportive comment. There is little reference so far on 24-hour RAR components in depression. Of these background, we would say the below previous references have a powerful impact on 24-hour RAR in middle aged and older people with depression. As our development of the present study is based on the studies, this background has been explained in the introduction (page 2, line 50-55).   

  1. Luik, A.I.; Zuurbier, L.A.; Direk, N.; Hofman, A.; Van Someren, E.J.W.; Tiemeier, H. 24-Hour Activity Rhythm and Sleep Disturbances in Depression and Anxiety: A Population-Based Study of Middle-Aged and Older Persons. Depression and anxiety 2015, 32, 684-692.
  2. Smagula, S.F.; Boudreau, R.M.; Stone, K.; Reynolds, C.F., 3rd; Bromberger, J.T.; Ancoli-Israel, S.; Dam, T.T.; Barrett-Connor, E.; Cauley, J.A.; Osteoporotic Fractures in Men Research, G. Latent activity rhythm disturbance sub-groups and longitudinal change in depression symptoms among older men. Chronobiology international 2015, 32, 1427-1437.

Comment 3: adding demographics to measurement is not appropriate.

Response to Comment 3: Demographics has been added into the section of “2.1. Participants” (page 2, line 87-92).

Comment 4: Sample items of the scale can be presented.

Response to Comment 4: Sample items of HDRS-17 have been added in the section (page 3, line 95-107)

Comment 5: Regression summary table should be integrated explain all predictors and outcomes.

Response to Comment 5: Thank you for a supportive comment. For binomial logistic regression analysis, the likelihood ratio test with forward selection has been applied to estimate regression models in table 3. In a case of the SPSS, statistics of all predictors are not typically outputted depending on a result of the likelihood ratio test with forward selection. Table 3 indicated integrated results of the likelihood ratio test with forward selection (page 7, Table 3).

Comment 6: Since the regression was the main point of objectives it should be prioritized.

Response to Comment 6: Thank you for a supportive comment. In the last paragraph of the introduction section (page 2, line 64-66) or the first paragraph of the discussion section (page 7, line 240), the main point of objective has been explained. 

Comment 7: Since the regression analysis impact should be highlighted presenting R square is very much needed.

Response to Comment 7: We applied the contribution rate of “Nagelkerke’s R square” to the estimated regression model. We have added description of a result regarding “Nagelkerke’s R square” in Data analysis or Result section, as well as the footnote of Table 3.

Comment 8: Conclusion is not sufficient.

Response to Comment 8: Thank you for a supportive comment. According to our results, the description of Conclusion has been revised carefully.

Comment 9: Please consider alphabetical order for the preparation of the list of reference.

Response to Comment 9: The list of references has been fixed carefully in Alphabetical order and numbering order.

Reviewer 2 Report

"24-hour rest-activity rhythm in middle-aged and older persons with depression" is an interesting study. Need minor clarification:

- The following demographic information were collected from electronic medical records: education, days of hospitalization, onset, presence of readmission, living with family, history of divorce, medication (antipsychotic or benzodiazepine), and independence of daily living activities according to the Barthel index.

In what way these information can be correlated to your study results?

- Provide abbreviation for UPSA-B during first time use (Line 114).

- Line 58 - .....this study focused on the follow points: Change to ".....this study focused on the following points:".

Author Response

Dear Reviewer2,

We would like to appreciate your supportive comments toward our paper. According to suggestion raised by each reviewer, our manuscript has been revised carefully as indicated in the following response to each reviewer’s comment. We believe the revised paper would provide novel findings to the readers of International Journal of Environmental Research and Public Health with Special Issue "Rehabilitation in Geriatric Psychiatry".

Comments and Suggestions for Authors

"24-hour rest-activity rhythm in middle-aged and older persons with depression" is an interesting study. Need minor clarification:

Comment 1 of Reviewer 2- The following demographic information were collected from electronic medical records: education, days of hospitalization, onset, presence of readmission, living with family, history of divorce, medication (antipsychotic or benzodiazepine), and independence of daily living activities according to the Barthel index. In what way this information can be correlated to your study results?

Response to Comment 1: Thank you for a supportive comment. As suggested by the reviewer, correlation between demographic data in patients with depression and 24-hour rest-activity rhythm (RAR) components seem to be important clinical viewpoint. However, as the present research limitation, the sample size of in- ore out-patients with depression (n = 29) was limited extremely and this issue got to be difficult to additionally analyze with demographic information including education, days of hospitalization, onset, presence of readmission, living with family, history of divorce, medication (antipsychotic or benzodiazepine), and independence of daily living activities as scored in the Barthel index. Considering this limitation carefully, we need to examine the issue in the next research. The description regarding this point has been added in the discussion section (page 9, line 318).  

Comment 2 of Reviewer 2- Provide abbreviation for UPSA-B during first time use (Line 114).

Response to Comment 2: We have described abbreviation for UPSA-B (page 3, Line 128)

Comment 3 of Reviewer 2- Line 58 - .....this study focused on the follow points: Change to ".....this study focused on the following points:".

Response to comment 3: Thank you for supportive comment. As suggested by the reviewer, we have revised the sentences (page 2, line 59).